# Salt Scaling Resistance of Variable w/c Ratio Air-Entrained Concretes Modified with Polycarboxylates as a Proper Consequence of Air Void System

**DOI:** 10.3390/ma15175839

**Published:** 2022-08-24

**Authors:** Aneta Nowak-Michta

**Affiliations:** Chair of Building Materials Engineering, Faculty of Civil Engineering, Cracow University of Technology, 24 Warszawska St., 31-155 Cracow, Poland; aneta.nowak@pk.edu.pl

**Keywords:** scaling resistance, water-cement ratio, superplasticizer, modified polycarboxylates, air-entrained concrete, air void system, air void parameters

## Abstract

The values of the air void parameters in hardened concrete (spacing factor L ≤ 0.200 mm and micro air content A_300_ ≥ 1.5%), determined on the basis of the Powers model, in concretes produced today do not always guarantee the frost resistance of the concrete, especially when in surface impact with the participation of de-icing agents. The literature indicates that the modified polycarboxylates used to liquefy concrete mixes are one of the factors involved in changing the air void system; therefore, the aim of the article was to determine the dependence of the air void parameters and the resistance to scaling of concretes liquefied to a constant consistency by the use of modified polycarboxylates in the spectrum of variability of the ratio w/c = 0.53 ÷ 0.30. In the research program, twelve concrete mixes were made with a constant proportion of aggregate and paste: six air-entrained—with a constant air content of 5.5 ± 0.5%—and six non-air-entrained. The air void parameters were determined in accordance with EN 480-11, while the resistance to scaling was determined in accordance with CEN/TS 12390-9 and assessed according to the criteria of SS 137244. The analysis of the test results showed that liquefaction with modified polycarboxylates did not affect the w/c limit values, enabling obtaining concretes resistant to scaling. They are, respectively, 0.35 in the non-air-entrained concretes and 0.50 in the air-entrained concretes with an air content of 5.5 ± 0.5. Moreover, the commonly used criterion for ensuring the frost resistance of air-entrained concretes, L ≤ 0.200 mm and A_300_ ≥ 1.5%, requires supplementing with the minimum value of the w/c ≤ 0.50.

## 1. Introduction

Scaling, which is a surface effect of frost with the participation of defrosting agents, is the dominant factor in the destruction of road and bridge concretes [1,2,3,4,5,6,7,8]. Defrosting agents used on road concrete pavements may cause several-millimeter splinters of hardened cement paste on the concrete surface; this is caused by an increase in osmotic pressure, microcracks due to thermal shocks, or chemical reactions [1,5,8,9,10,11]. While freezing on a concrete surface, saline solution forms a two-material composite of ice and concrete. The temperature of this composite drops below the melting point of the solution. Then, the ice layer shrinks five times more than the concrete underlying it does [2,3,8]. As a result, the top layer of concrete breaks into many small sheets. Tensile stress develops in concrete along the perimeter of these patches, causing the defects to spread over the concrete surface, culminating in the removal of a thin layer of concrete. Fracture mechanics allows us to predict that the fracture propagation and the depth at which the fracture rotates parallel to the transition zone are dictated by the mechanical properties of the constituent materials. Therefore, if the properties of the concrete do not differ significantly across the surface, each freeze–thaw cycle causes a relatively constant amount of damage. The resistance to scaling of concrete is determined by the strength of the top layer. The estimated chipping stress is marginal in relation to the tensile strength of the cement binder (∼3 MPa) [2]. Any treatments or activities that may weaken the surface, including bleeding, result in more scaling [2,5,6,8,12,13].

Resistance to scaling depends primarily on the degree of concrete saturation with water and the pore microstructure in the cement paste. The pore microstructure in the cement paste is shaped by air entrainment (AE) and an appropriately low value of the water–cement ratio (w/c), which will minimize the presence of capillary pores [1,2,3,4,5,6,7,8,14,15,16,17,18,19,20,21,22]. According to ACI 201.2R [7], the water–cement ratio in frost-resistant concrete should not exceed 0.50, and in the case of thin sections, i.e., bridge slabs and curbs, it should not exceed 0.45.

The w/c ratio is the most important parameter of all concrete mixtures; it is responsible for the material properties and determines the durability and strength [1,5,6,7,8]. In particular, lowering the w/c value reduces the susceptibility to bleeding, increases the strength of the top layer of concrete, and, consequently, increases the resistance to scaling [23]. Concretes with a w/c ≤ 0.30 do not require air entrainment to ensure resistance to scaling [1,5,6]. The bleeding of cement pastes with w/c = 0.3 is very small and comparable to the 20% air-entrained cement paste with w/c = 0.45. The strength of concretes with w/c ≤ 0.3 is about 50–60 MPa. Therefore, low w/c concretes do not require air entraining to prevent scaling as they have negligible bleeding and a high strength of the top layer, which is not different from the strength of the entire volume of concrete. However, properly air-entrained concretes (ACs), in order to ensure resistance to scaling, should have a water–cement ratio not exceeding 0.5 [1,5,7,8]. At constant air content, the spacing factor (L) depends on the w/c ratio. Due to the aggregation of fine air bubbles, as the w/c value increases, the spacing factor increases, and the specific surface area (α) decreases [3,4].

The aspect of ensuring durability is the dominant factor of concretes produced today [18,24,25,26,27,28]. However, the current commonly used modifications of concrete compositions with admixtures [28], additives [18,27], modern calcium sulfoaluminate cement [26], or geopolymer [24,25] binders significantly change the structure of the concrete. Consequently, the values of the air void parameters (AVPs) in hardened concrete, L ≤ 0.200 mm and micro air content (A_300_) ≥ 1.8%, determined on the basis of the Powers model [6,7,9], do not always guarantee concrete frost resistance, especially in the case of surface interaction with defrosting agents, as is shown in the literature [1,3,4,7,29,30,31,32] and the author’s own research [33,34] on concrete produced today.

One of the significant problems is the influence of the side effect of the superplasticizer (SP) in the form of additional air entrainment, which influences the air void structure (AVS) and its relationship with the concrete frost resistance. The problem especially concerns self-compacting concretes with a very complex system of the compatibility of the admixtures used, as well as the rheological issues [15,18,22,28].

In the previously realized research program [33] concerning the influence of the SP on the modified polycarboxylate (MP) base on the resistance to scaling of ordinary concretes, and its evaluation in the light of the obtained AVP, a constant value of w/c = 0.45 was adopted. The variable parameter was the SP content, which was determined to obtain the S1-S4 consistency class [35]. In the air-entrained mixtures, the air-entraining admixture content was constant, and it was established for the mixture without the SP, in which the air content was 6%. Due to the fact that all the air-entrained concretes (ACs), after 56 cycles of scaling, showed a very good resistance, no effect of the SP on their resistance to scaling was observed. In non-air-entrained concretes (NACs), with the increase in the SP content and the simultaneous liquefaction of the NAC, the resistance to scaling decreased as a result of the weakening of the top layer of concrete. Moreover, the SP significantly changed the AVP, especially in the AC, causing a significant decrease in the α, which resulted in an increase in the L and a decrease in the A_300_. It was also confirmed that the AVP takes into account both the air-entrainment pores and the pores that are a side effect of the SP operation. As a consequence, it was found that SP shapes AVP, the values of which, as standard values in concrete for the assessment of frost resistance, L ≤ 0.200 mm and A_300_ ≥ 1.5%, are also not correlated with the tested frost resistance in the case of ordinary concretes.

In the research program [36] on the scope of the influence of w/c on the AVS, it was shown that the AE of concrete mixtures at the level of 5.5 ± 0.5% allows the obtaining of the AVP required to ensure frost resistance [6,14]: L ≤ 0.200 mm and A_300_ ≥ 1.50%. In addition, the AVP is described by mathematical relations with high correlation coefficients: total air content (A), L, α, and A_300_ in both AC and NAC change with the w/c ratio. Lowering the w/c ratio value increases the viscosity of the concrete mixes, thus influencing the air-entraining process; at a constant consistency, this requires a higher dose of SP, the side effect of which is an additional content of pores above 300 µm, which are unfavorable to frost. The w/c ratio does not affect the content of pores below 300 μm, which are favorable for frost resistance. The most favorable structure of air entrainment was obtained in concrete with w/c = 0.45 due to the optimal viscosity of the concrete mixture allowing for the emergence of large pores during compaction and preventing the coalescence of fine pores, as well as to the content of SP, which does not cause additional AE.

In the previously performed test program [33], at a constant value of the w/c ratio = 0.45, a very good resistance to scaling of all the AC was obtained, which made it impossible to determine the relationship between AVP and the resistance to scaling. Moreover, the research on the influence of w/c on the AVS in concrete [36] shows that as the w/c decreases and the demand for SP increases, the value of the concrete, unfavorable for frost resistance, increases. Therefore, the aim of the article was to determine the relationship between AVP and the resistance to scaling of concretes liquefied to a constant consistency with the use of modified polycarboxylates in a wide range of the variability of the w/c ratio.

## 2. Materials and Methods

### 2.1. Materials

#### 2.1.1. Cement

CEM I 42.5R cement was used in the concrete mixes, meeting the requirements for reference concrete according to EN 480-1 [37]. The cement has a specific surface according to a Blaine of 4516 cm^2^/g and a density of 3.1 g/cm^3^. The physical and chemical properties of the cement specified by the manufacturer are presented in Table 1.

#### 2.1.2. Aggregates

Natural sand and a crushed basalt aggregate with a maximum grain of 16 mm and a sand point of 36.8% were used in the concrete mixes. The crumb pile was iterated to obtain an aggregate mix with a minimum sum of cavity and water demand. Based on our own analysis of the grain size distribution of individual aggregates, a grain size distribution curve was constructed for the crumb pile, composed by iteration. The graining curve of the aggregate used, presented in Figure 1, has been drawn into the boundary curves for the reference concrete in accordance with EN 480-1 [37]. The obtained graining curve shows insignificant contents of oversize and undersize. However, they were allowed in the test program due to the fulfillment of more important criteria in the scope of tightness and water demand.

#### 2.1.3. Chemical Admixtures

Table 2 lists the properties of the air-entraining admixture (AEA) and the superplasticizer (SP), as specified by the manufacturers. The chemical compositions of the admixtures are protected by the manufacturer’s patent. The percentage contents of the AEA and SP were calculated in relation to the cement weight. The amounts of AEA and SP were determined experimentally in order to obtain a constant air content (Vp) in the air-entrained mixes at the level of 5.5 ± 0.5% and the consistency class of S2 for all the concrete mixes (slump 50–90 mm) [35].

### 2.2. Mix Proportion and Its Preparation

Twelve concrete mixes were made in order to unequivocally determine the relationship between the AVP and the resistance to scaling in the concretes liquefied with SP based on MP. Six air-entrained concretes (ACs), labeled AC1–AC6, and six non-air-entrained concretes (NACs), labeled NAC1-NAC6, were performed. In order to minimize the influence of factors shaping the AVS, the following constant parameters were adopted: the paste content of 30%; type, grain size, and aggregate content (sand P0/2 = 717 kg/m^3^ and basalt grit G4/8 = 459 kg/m^3^ and G8/16 = 688 kg/m^3^); and cement type CEM I 42.5R, as well as the consistency of the concrete mixes (slump 50 ± 10 mm) and air content in the concrete mixes (Vp). In the research program, the composition of the paste was variable in terms of the w/c, content of cement, water, SP, and AEA. To isolate the variability of the scaling resistance, the w/c varying in the range of 0.53 ÷ 0.30 was used in the pavement concretes (Table 3). NAC1 non-air-entrained control concrete fulfilled the requirements for the reference concrete III according to EN 480-1 [37] and, similarly to the air-entrained concrete of the same AC1 formulation, did not contain SP. The remaining concrete mixes, both the air-entrained AC2–AC6 and the non-air-entrained NAC2–NAC6, were liquefied with SP based on MP. The basic recipes of the mixes with the same numbers 1 to 6 differed only as to the content of the air-entraining admixture (AEA). One AEA type, the base of which is a combination of natural resins and synthetic tensides, was used in all the air-entrained mixtures, AC1–AC6. The amount of AEA was regulated to obtain Vp at the level of 5.5 ± 0.5%.

The mixes were made in a horizontal plan mixer with a volume of 100 dm^3^. The dry ingredients, the aggregates and the cement, were mixed for 0.5 min; then, part of the mixing water was added and mixed for another 0.5 min. The SP was added successively, along with 2 dm^3^ of water and mixed for 1 minute. In the air-entrained mixes, as the last step, AEA was added with 2 dm^3^ of water and then mixed for 2 min. The total mixing time for the NAC mixes was 2 min, and for the AC mixes, it was 4 min. The test samples were demolded after 24 h and stored in a chamber at 20 ± 2 °C and a humidity of 95 ± 5%, in accordance with EN 12390-3 [38].

### 2.3. Concrete Tests

The test scheme of the concrete mixes and hardened concretes is presented in Figure 2.

#### 2.3.1. Concrete Mix Tests

The concrete mixes were tested within 10 minutes from the mixing of the components. The consistency, by the slump-test method, in accordance with EN 12350-2 [39]; the density, in accordance with EN 12350-6 [40]; and the air content, by the pressure method, in accordance with EN 12350-7 [41], were determined.

#### 2.3.2. Hardened Concrete Tests

Concrete tests were carried out after 28 days of curing in a chamber with a temperature of 20 ± 2 °C and a humidity of 95 ± 5%, in accordance with EN 12390-3 [38]. The scaling resistance according to CEN/TS 12390-9 [42] and the air void parameters according to EN 480-11 [43] were determined. The compressive strength (f_c_) according to EN 12390-3 [38] was determined for three cubes with a side length of 150 mm, and a density (D) in accordance with EN 12390-7 [44] was determined for five cubes with a side of 150 mm.

##### Scaling Test

The scaling resistance tests were carried out for each series of concrete on four samples with dimensions of 140 mm × 72 mm × 50 mm, in accordance with CEN/TS 12390-9:2017 Testing hardened concrete. Freeze-thaw resistance with de-icing salts—Scaling [42]. The samples were subjected to 56 cycles of scaling in the presence of 3% NaCl in an automatic freezing and thawing chamber. Mass losses due to scaling were determined every 7 cycles. The scaling resistance was assessed according to the criteria of the Swedish standard SS 137244: 2019 Concrete testing—Hardened concrete—Scaling at freezing [45]. The standard specifies the concrete quality categories depending on the mass of scaling after 28 (m_28_), 56 (m_56_), and 112 (m_112_) cycles, as presented in Table 4.

##### Air Void Parameters (AVP)

The AVPs were determined in accordance with EN 480-11 [43] for each series of hardened concretes in two samples with dimensions of 150 mm × 150 mm × 20 mm, using the automatic RapidAir 457 system. As a result of the analysis, our basic AVPs in hardened concrete were determined: A—total content of air, α—specific surface, L—spacing factor, A_300_—micro air content.

## 3. Results

The results of the concrete mixtures and the compressive strength (f_c_) and density (D) of the hardened concretes after 28 days of curing are presented in Table 5.

### 3.1. The Results of Concrete Mix Tests

The slumps of all the mixtures were in the range of 50–90 mm and according to EN 206 [35], and they can be classified as consistency class S2 according to the adopted assumption (Figure 3).

The densities (D) of the NAC mixtures ranged from 2380 to 2443 kg/m^3^, while the densities of the AC mixtures ranged from 2283 to 2352 kg/m^3^. The air contents (Vp) ranged from 0.8 to 3.0% in the NAC mixtures and from 5.0 to 6.0% in the AC mixtures. The Vp in the AC mixtures met the assumed criterion of 5.5 ± 0.5%. The obtained values confirm the adopted assumptions for the dosing of the SP and AEA admixtures, which enable the assumed consistency class and air content in the AC to be achieved.

### 3.2. The Results of Hardened Concretes Tests

The compressive strengths ranged from 48.0 to 82.8 Mpa in the AC and from 31.1 to 59.7 Mpa in the NAC. The compressive strength of both the NAC and the AC concretes increases with the reduction of the w/c value (Figure 4). In the NAC concretes, it is a linear relationship, while in the AC concretes the linear relationship was disturbed by fluctuations in the air content of V_p_ = 5.5 ± 0.5%. The strength drops in the AC in relation to the NAC are at the level of 19–38% and depend on both the Vp and the w/c ratio. The densities ranged from 2376 to 2441 kg/m^3^ in the AC and from 2228 to 2363 kg/m^3^ in the NAC. The densities of both the NAC and the AC increase as the w/c values decrease.

#### 3.2.1. Salt Scaling

The losses of mass after 7, 14, 21, 28, 35, 42, 49, and 56 cycles and the concrete qualities after 56 scaling cycles, according to the SS 137244 criteria [45], are shown in Table 6. The mass losses due to scaling decrease as the value of the w/c ratio decreases in both the AC and the NAC. The AC showed a much higher resistance to scaling than the NAC. NAC1-NAC4 are of unacceptable quality, while NAC5-NAC6 are of acceptable quality. AC1 is unacceptable; AC2 and AC3 are acceptable; and AC4 is good, while AC5 and AC6 are of very good quality.

#### 3.2.2. The Air Void Parameters

The AVP parameters, A, L, α, and A_300_, characterizing the air void system in hardened concretes, are presented in Table 7.

## 4. Discussion

### 4.1. The Influence of w/c on the Scaling Resistance

Mass losses due to scaling (Figure 5) appear after just 7 cycles in the non-air-entrained concretes, with w/c ≤ 0.5 (NAC1; NAC2). However, they do not exceed 0.1 kg/m^2^, and thus, the concretes are of very good quality [45]. After 14 cycles, the NAC1 concrete (w/c = 0.53) is of acceptable quality, and the NAC2 (w/c = 0.5) is of good quality. After 21 cycles, the non-air-entrained concretes with w/c ≤ 0.5 (NAC1; NAC2) are of unacceptable quality, and the NAC3 concrete is of acceptable quality. Between the 21st and the 28th cycle, the NAC1–3 concrete samples (w/c ≤ 0.45) disintegrate, i.e., their volume is destroyed [2,3,5,6]. After 28 cycles, the first flaking appears in the NAC4 concrete (w/c = 0.4). Between the 28th and the 35th cycle, the remaining non-air-entrained NAC4–6 concretes and the control air-entrained concrete AC1 (w/c = 0.53) are scaled off. Between the 35th and the 42nd cycle, the AC1 concrete is destroyed, further scaling occurs in the NAC4-6 concretes, and the first flaking appears in the AC2 and AC3 concretes. Between cycle 42 and 49, the NAC4 concrete becomes of unacceptable quality, while the NAC5–6 and AC2–3 concretes are of acceptable quality and remain in these categories until cycle 56. Only concretes AC4–6 (w/c ≤ 0.4) do not show any signs of flaking after 56 cycles and remain of very good quality.

The dependences of mass loss and w/c after 56 scaling cycles for both the AC and the NAC presented in Figure 6 indicate evident decreases in mass losses along with the reduction of the w/c value. The variability of the w/c ratio determined in the test program made it possible to obtain AC concretes with variable resistance to scaling from unacceptable to very good quality, which makes it possible to assess the influence of SP on the resistance to scaling. This type of analysis was not possible in [33] due to the obtained very good quality of all the air-entrained concretes. Of course, we are dealing here with a simultaneous interaction of three changing parameters, i.e., the w/c ratio, the cement content, and the increasing SP content as the w/c decreases. The limit value in the NAC of acceptable quality is w/c = 0.35, while in the AEC with an air content of 5.5 ± 0.5% it is w/c = 0.5. The obtained results confirm the literature data [1,5,7,8,46] with regard to the w/c limit values enabling the obtaining of concretes resistant to scaling; in addition, they confirm the same MP with which the new generation of concrete mixes is liquefied; although they change the air void structure [36], they do not affect the resistance to scaling. Thus, the problem of the decrease in resistance to scaling in the SCC concretes [28], as a result of additional air entrainment when liquefied with modified polycarboxylates, does not apply to pavement concretes, in which concrete mixes with consistency classes S2–S4 are standardly used [35].

The dependences of the mass losses after 56 scaling cycles and the compressive strength of the concrete after 28 days in the NAC, as presented in Figure 7, show an evident increase in resistance to scaling with increasing strength. However, in the AC, due to the dominant influence of air entrainment on both the scaling resistance and the decrease in compressive strength resulting from it, this relation has not been confirmed in the tests.

### 4.2. Assessment of Resistance to Scaling in the Light of AVP

Figure 8, Figure 9, Figure 10 and Figure 11 show the AVP and w/c dependencies together with the assessment of the quality of the concrete after 56 scaling cycles.

The total air contents in the NAC both at the stage of concrete mixes, Vp—Table 5, and the hardened concretes, A—Table 7 and Figure 8, increased from 0.8 to 2.4%, as compared to the control concrete NAC1, which was the only one that was not liquefied with SP. Therefore, Ref. [33] the side effect in the form of additional air entrainment after adding SP based on modified polycarboxylates is confirmed. A synergistic air-entraining effect of AEA and SP is observed in the AC. As shown in Figure 8 presenting the A and w/c relation, the resistance to scaling in the AC at a constant air-entrainment level significantly depends on the w/c ratio. For concretes with w/c = 0.53, the air entrainment at the level of 7.4% is not sufficient to ensure resistance to 56 scaling cycles. It should be noted that in order to obtain a constant consistency, in both the NAC and the AC concretes, the SP dose was increased, which weakened the top layer of concrete and, consequently, reduced the resistance to scaling [33].

The specific surfaces of all the NACs are below the recommended value of 24 mm^−1^ [7] for the frost resistance, and the AC is above the recommended value. In both the AC and the NAC, the specific surface decreases with a decrease in w/c (Figure 9), despite the fact that in the NAC the air content increases with a decrease in w/c. Additionally, according to the literature data [7,14,15,16], the frost resistance should increase, at a constant air content, with a relatively finer structure of air entrainment in the concrete. It should be emphasized here that the decrease in the specific surface area of the pores with the decrease in w/c results from the increasing content of SP based on MP, which causes a disturbance in the AVS. At a constant air content, the content of the preferred pores (entrained air) decreases, while the that of the unfavorable pores (entrapped air) increases [36]. However, it has no effect on the resistance to scaling, for which, as shown by the test results, the strength of the top layer is more important and is mainly determined by the value of the w/c ratio.

All the NAC1–6 concretes have L > 0.292 mm, while all the AC1–6 concretes have L ≤ 0.200 mm (Figure 10). That is, all the ACs at the level of 5.5 ÷ 0.5% meet the commonly accepted criterion for frost-resistant concretes, and yet, their resistance to scaling [7,9], with similar values of L = (0.138 ÷ 0.165) in the AC2-6 concretes, increases with the lowering of the w/c value. Attention should be paid to concrete AC1, which, despite obtaining the value of L = 0.101 mm, showed unacceptable quality after 56 scaling cycles.

All the AC1–6 air-entrained concretes had micropore contents higher than recommended for frost resistance [7], exceeding 1.5%. In the NAC, the values did not exceed 1.5%. However, in the case of the AC, trends similar to those for the parameters L and α can be observed (Figure 11). Despite the fulfilment of the criterion, AC1 concrete with w/c = 0.53 is of unacceptable quality. Moreover, with a comparable value of A_300_ = (2.35 ÷ 3.05), the resistance to scaling increases with the reduction of the w/c value. The obtained AVP values clearly indicate the influence of the modified polycarboxylates used to liquefy concrete mixes on the air void system. Nevertheless, by disturbing the air void system and deteriorating its parameters, they do not affect the resistance to scaling. Thus, in the light of the obtained results, it is necessary to look at the evaluation of the resistance to scaling in a new way, taking into consideration both the AVP and the w/c ratio.

The AVP obtained for AC1 concrete with w/c = 0.53, L = 0.101 mm, and A_300_ = 4.28%, despite meeting the commonly used criterion for frost resistance assessment, L ≤ 0.200 mm and A_300_ ≥ 1.5% [6,7,9], is concrete of unacceptable quality in the light of the requirements of SS 137244 [45]. Therefore, the above requirement, frequently [28,29] used spontaneously to assess frost resistance in the light of the obtained results, indicates that it is not spontaneously sufficient to ensure concrete scaling resistance. It should be noted that the AC1 concrete was not liquefied with SP; so, we are not dealing here with the side effect of air-entraining MP. The obtained test results confirm the inability to determine the unambiguous relationship between AVP and the resistance to scaling. In air-entrained concrete, it is necessary to supplement these requirements of L ≤ 0.200 mm and A_300_ ≥ 1.5% with the minimum value of w/c ≤ 0.5. In addition, it should be emphasized that the requirement applies only to concretes made with Portland cement without additives and admixtures other than AEA and SP based on modified polycarboxylates.

## 5. Conclusions

The conducted research program, the aim of which was to determine the relationship between AVP and the resistance to scaling of concretes liquefied to a constant consistency with the use of MP in a wide range of variability of the w/c ratio, allowed for the formulation of the following conclusions:The w/c variability in the range of 0.53 ÷ 0.30 determined in the test program made it possible to obtain AC with variable resistance to scaling from unacceptable to very good.Liquefaction with MP did not affect the w/c limit values, enabling the obtaining of concretes resistant to scaling. They are, respectively, 0.35 in NAC and 0.50 in AC of Vp = 5.5 ± 0.5.Resistance to scaling in concrete modified with polycarboxylates grows with a decrease in w/c. A more significant effect can be observed in NAC, because air entrainment is an equally important factor in AC.MPs changed the AVS of AC; however, they did not affect their resistance to scaling. As a result of an increase in the SP content, at a constant air content, with a decrease in w/c in both the NAC and the AC, the entrained-air content decreases against the unfavorable entrapped air.The commonly used criterion for ensuring the frost resistance of AC in the range of AVP, L ≤ 0.200 mm and A_300_ ≥ 1.5%, requires supplementing with the minimum value of the w/c ≤ 0.50.

## Figures and Tables

**Figure 1 materials-15-05839-f001:**
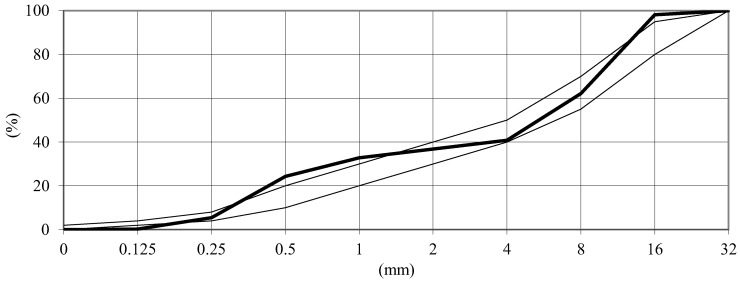
The grain size distribution of aggregate.

**Figure 2 materials-15-05839-f002:**
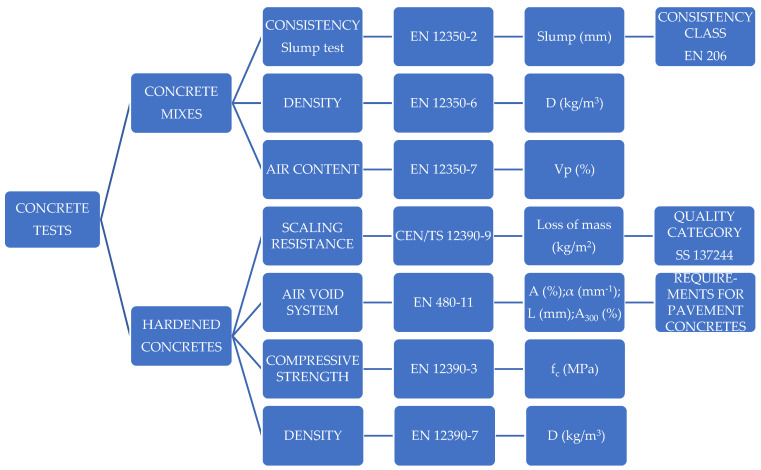
Test scheme.

**Figure 3 materials-15-05839-f003:**
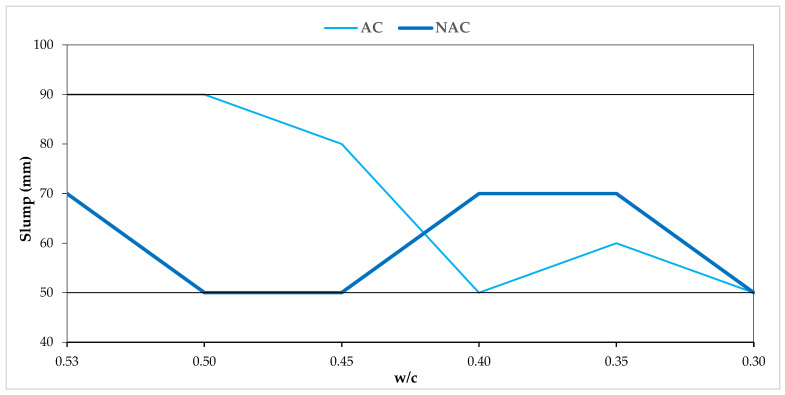
Slump and w/c relation in AC and NAC.

**Figure 4 materials-15-05839-f004:**
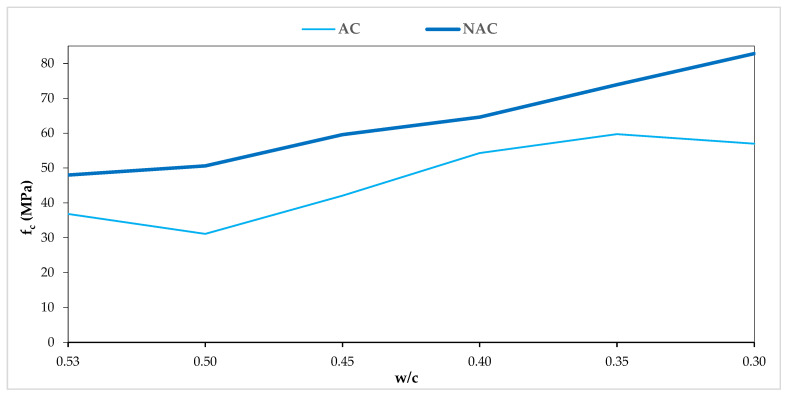
f_c_ and w/c relation in AC and NAC.

**Figure 5 materials-15-05839-f005:**
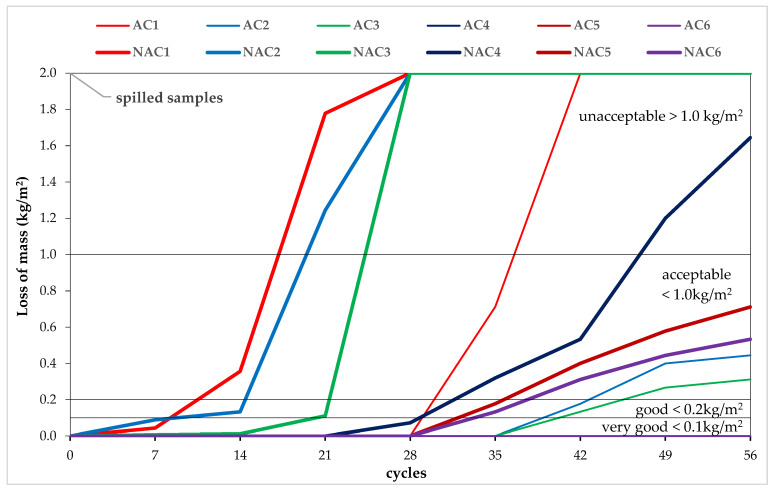
Losses of masses after 7, 14, 21, 28, 35, 42, 49, 56 cycles of scaling.

**Figure 6 materials-15-05839-f006:**
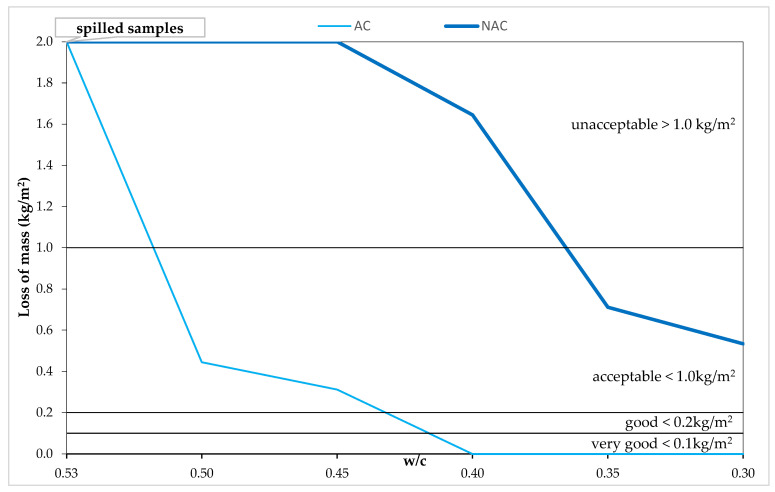
Mass losses after 56 cycles of scaling and w/c relation in AC and NAC.

**Figure 7 materials-15-05839-f007:**
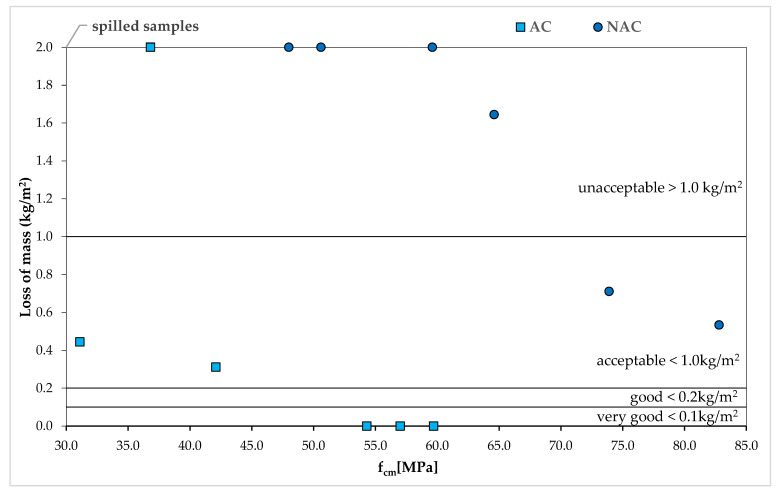
Mass losses after 56 cycles of scaling and f_cm_ relation in AC and NAC.

**Figure 8 materials-15-05839-f008:**
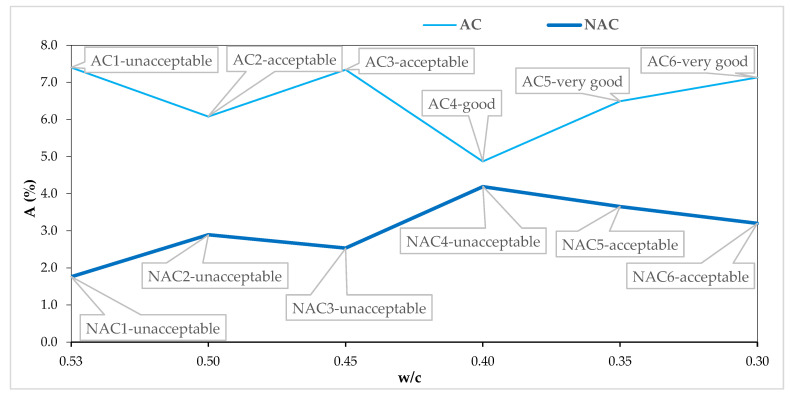
A and w/c relation in AC and NAC.

**Figure 9 materials-15-05839-f009:**
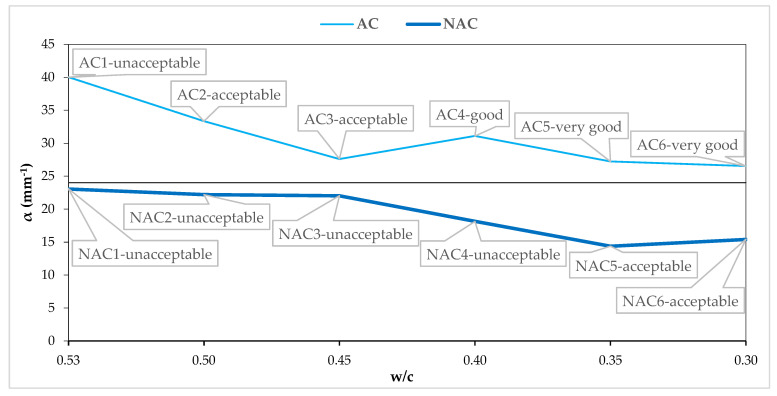
α and w/c relation in AC and NAC.

**Figure 10 materials-15-05839-f010:**
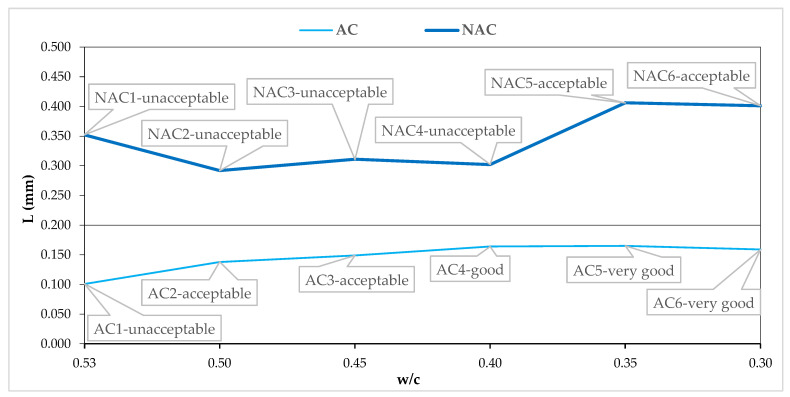
L and w/c relation in AC and NAC.

**Figure 11 materials-15-05839-f011:**
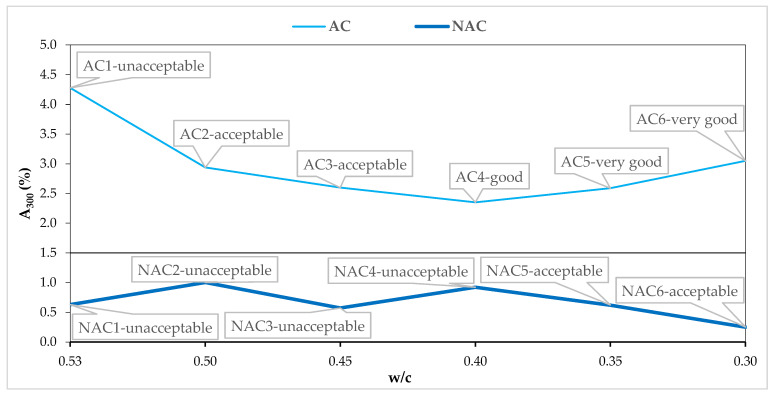
A_300_ and w/c relation in AC and NAC.

**Table 1 materials-15-05839-t001:** The physical and chemical properties of CEM I 42.5R.

Setting Time, Vicat Test (min)		Water Demand (%)	Compressive Strength (MPa)	Chemical Analyses (%)		Insoluble Residue (%)	Loss on Ignition (%)
Initial	Final			SO_3_	Cl		
172	224	28.7	60.7	3.02	0.048	0.55	3.47

**Table 2 materials-15-05839-t002:** The properties of admixtures.

Property	AEA	SP
Main base	Combination of natural resins and synthetic surfactants	PCE
Specific gravity at 20 °C (g/cm^3^)	1.01	1.07 ± 0.02
pH value at 20 °C	11.0	4.7 ± 1
Chloride ion content (% mass)	≤0.1	<0.1
Alkali content (Na_2_O_eqiv._) (% mass)	<0.8	≤0.6

**Table 3 materials-15-05839-t003:** The components of concretes.

Symbol of Concrete	CEM I 42.5R (kg/m^3^)	w/c	SP	AEA	Paste Volume (%)	Sand 0/2 mm (kg/m^3^)	Gravel 2/8 mm (kg/m^3^)	Gravel 8/16 mm (kg/m^3^)
			(kg/m^3^)					
NAC1	350	0.53	-	-	30	717	458	688
AC1	350	0.53	-	1.05	30	717	458	688
NAC2	360	0.50	0.36	-	30	717	458	688
AC2	360	0.50	0.36	1.08	30	717	458	688
NAC3	383	0.45	0.57	-	30	717	458	688
AC3	383	0.45	0.57	1.15	30	717	458	688
NAC4	409	0.40	1.02	-	30	717	458	688
AC4	409	0.40	0.82	1.02	30	717	458	688
NAC5	439	0.35	1.76		30	717	458	688
AC5	439	0.35	1.54	1.10	30	717	458	688
NAC6	474	0.30	2.84		30	717	458	688
AC6	474	0.30	2.37	1.42	30	717	458	688

**Table 4 materials-15-05839-t004:** Category of scaling resistance in presence of 3% NaCl [45].

Concrete Quality	m_56_ (kg/m^2^)	m_56_/m_28_	m_112_ (kg/m^2^)
very good	<0.1	-	-
good	<0.2	-	-
	<0.5	<2	-
	-	-	<0.5
acceptable	<1.0	<2	-
	-	-	<1.0
unacceptable	>1.0	>2	-
	-	-	>1.0

**Table 5 materials-15-05839-t005:** Test results of concrete mixtures and hardened concretes.

Symbol of Concrete	Concrete Mixes			Hardened Concretes	
	Slump (mm)	D (kg/m^3^)	Vp (%)	f_c_ (MPa)	D (kg/m^3^)
NAC1	70	2383	0.8	48.0	2376
AC1	90	2283	5.8	36.8	2316
NAC2	50	2388	1.7	50.6	2408
AC2	90	2288	5.0	31.1	2228
NAC3	50	2390	2.0	59.6	2441
AC3	80	2295	5.5	42.1	2337
NAC4	70	2380	3.0	64.6	2437
AC4	50	2315	5.0	54.3	2353
NAC5	70	2380	2.6	73.9	2434
AC5	60	2352	5.4	59.7	2350
NAC6	50	2443	2.4	82.8	2377
AC6	50	2305	6.0	57.0	2363

**Table 6 materials-15-05839-t006:** The decrease in mass (kg/m^2^) after scaling cycles and the quality of concretes after 56 cycles of de-icing salt scaling according to SS 137244 [45].

Symbol of Concrete	7 Cycles	14 Cycles	21 Cycles	28 Cycles	35 Cycles	42 Cycles	49 Cycles	56 Cycles	The Quality of Concretes
NAC1	0.04	0.36	1.78	>1 *	>1 *	>1 *	>1 *	>1 *	unacceptable
AC1	0	0	0	0	0.71	>1 *	>1 *	>1 *	unacceptable
NAC2	0.09	0.13	>1 *	>1 *	>1 *	>1 *	>1 *	>1 *	unacceptable
AC2	0	0	0	0	0	0.18	0.40	0.44	acceptable
NAC3	0.09	0.18	0.53	>1 *	>1 *	>1 *	>1 *	>1 *	unacceptable
AC3	0	0	0	0	0	0.13	0.27	0.31	acceptable
NAC4	0	0	0	0.07	0.32	0.53	>1 *	>1 *	unacceptable
AC4	0	0	0	0	0	0	0	0	good
NAC5	0	0	0	0	0.18	0.40	0.58	0.71	acceptable
AC5	0	0	0	0	0	0	0	0	very good
NAC6	0	0	0	0	0.13	0.31	0.44	0.53	acceptable
AC6	0	0	0	0	0	0	0	0	very good

* samples were disintegrated.

**Table 7 materials-15-05839-t007:** AVP.

Symbol of Concrete	A (%)	L (mm)	α (mm^−1^)	A_300_ (%)
NAC1	1.76	0.352	23.06	0.63
AC1	7.40	0.101	40.02	4.28
NAC2	2.89	0.292	22.21	1.00
AC2	6.08	0.138	33.36	2.94
NAC3	2.53	0.311	22.01	0.57
AC3	7.34	0.149	27.60	2.60
NAC4	4.19	0.302	18.15	0.92
AC4	4.87	0.164	31.10	2.35
NAC5	3.65	0.406	14.36	0.62
AC5	6.49	0.165	27.24	2.59
NAC6	3.20	0.401	15.40	0.25
AC6	7.13	0.159	26.54	3.05

## Data Availability

Not applicable.

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
