# Peer review of "Salt Scaling Resistance of Variable w/c Ratio Air-Entrained Concretes Modified with Polycarboxylates as a Proper Consequence of Air Void System"

_materials, 2022, doi:10.3390/ma15175839_

Round 1
Reviewer 1 Report
The paper presents a good topic related to. Salt scaling resistance of variable w/c ratio air-entrained concretes modified with polycarboxylates as a proper consequence of Air Void System. The following attached file to improve the paper before publication.

Reviewer 2 Report
Dear Reviewer,
Your work is interesting, however, a lot more work is available in this area. What is the novelty of this study? The following points need to be incorporated to enhance the quality of paper.
What is the power model, what are its limitations?
Line 67: define Vp, also all abbreviations and symbols need to be defined as their first appearance.
What does A300 mean in abstract first line.
use short sentences while writing. Long sentences confuse the statement. Line 80-85 a single statement, reduce it in short one.
Figure 1. The grain size distribution of aggregate. The grain size distribution is crossing outside the boundary curves from 0-.126; 0.3 -1.2; 15-32. is it allowed by EN 480-1 code?
Line 159-165; tabular comparison will be easier to understand.
A flow chart can be drawn mentioning tests name and relative standard for better comprehension.
Heading 3.2 use hardened concrete instead of only concretes.
Table 4 and 5 can be made together. for linking fresh and hardened properties test.
Table 7 can be added with Table 6. Concrete quality can be mentioned in table 6. A remarks/ reasoning column can be added as no reason is provided in heading 3.2.1.
heading 4.1 critical discussion and reasoning is missing. It's just reporting the results which are already known in this area. also compare your results from other studies as well.
Heading 4 graphs can be redrawn more Betterly.
Conclusions need to be rewritten.
Reviewer 3 Report
Salt scaling resistance of variable w/c ratio air-entrained concretes modified with polycarboxylates as a proper consequence of Air Void System
Abstract need revision with some quantitative results.
Some more latest studies are required in the introduction section to further highlight the importance of this study. For example:
1. Kubissa, W. (2020). Air permeability of air-entrained hybrid concrete containing CSA cement. Buildings, 10(7), 119.
2. Hodul, J., Žižková, N., & Borg, R. P. (2020). The influence of crystalline admixtures on the properties and microstructure of mortar containing by-products. Buildings, 10(9), 146.
Table 1, how properties were obtained?
Figure 1, how grain size distribution was obtained?
Section 2.2, how concrete mixes were selected, proper reasons are required.
Section 3.1, there is a need to include figure of slump test as well. Further results are not compared with the previous studies.
Section 3.2, the failure modes of compressions tests are required to present along with figures and proper disuccions.
Also, Conclusions are too limited to proof the significant outcome of this study.
Round 2
Reviewer 1 Report
The author has addressed all the comments of the reviewers. Therefore, the revised manuscript can be accepted for publication in J. Materials
Reviewer 2 Report
paper is much improved and can be accepted.